


# Does peatland rewetting mitigate extreme rainfall events?
Shirin Karimi[a], Eliza Maher Hasselquist[a], Järvi Järveoja[a], Virginia Mosquera[a], and Hjalmar Laudon[a]
[a] Swedish University of Agricutural Sciences, Department of Forest Ecology and Management, Umeå,
Sweden
*Corresponding author: shirin.karimi@slu.se
**Abstract**
Pristine peatlands are believed to play an important role in regulating hydrological extremes because
they can act as reservoirs for rainwater and release it gradually during dry periods. Therefore, rewetting
of drained peatlands is considered an important strategy to reduce the catastrophic effects of flooding.
With the anticipation of more frequent extreme rainfall events due to a changing global climate, the
importance of peatland rewetting in flood mitigation becomes even more important. To date, empirical
data showing that rewetting actually restores the hydrological function of drained peatlands is largely
lacking, particularly in Sweden. To assess whether rewetting peatlands can mitigate extreme rainfall
events and ensure water security in a future climate, we measured event-based runoff responses before
and after rewetting using a BACI approach (before-after and control-impact) within a replicated,
catchment scale study at the Trollberget Experimental Area in northern Sweden. High-resolution
hydrological field observations, including groundwater table level, discharge, and rainfall data were
collected over four years, allowing us to detect and analyze 17 rainfall-runoff events before and 30
events after rewetting. Our rainfall-runoff analysis revealed that rewetting significantly decreased peak
flow, runoff coefficient, and reduced the overall flashiness of hydrographs, making the rewetted site
function more like the pristine control peatland. However, "lag time" which was already similar to
pristine conditions was pushed farther away from pristine conditions following rewetting. We found
that the rewetted site experienced an increase in the groundwater table level following rewetting and
this was consistently observed across all distances from the blocked ditch within the peatland, providing
complementary data for our event-based analysis. In summary, our findings suggest that peatland
rewetting has the potential to mitigate flood responses, however, further research over a longer time
period is needed as peat properties and the peatland vegetation will develop and change over time.
**Keywords**
**Boreal landscape, peatland hydrology, rewetting, flood mitigation**



# 1. Introduction

Peatlands are the predominant wetland type in the boreal biome. They encompass 15% of the boreal region and serve as significant carbon sinks and methane sources, playing a crucial role in regulating the global climate (Helbig et al., 2020). In recent years, there has been an increased recognition of the importance of peatlands in carbon capture, flood management, water quality, and biodiversity (Holden et al., 2017). Regrettably, these valuable ecosystems have undergone substantial human-induced damages, with more than half of the peatlands in Europe estimated to have been lost through drainage for agriculture, forestry, or peat extraction (Andersen et al., 2017). Drained peatlands cannot sustain critical ecosystem services, imposing a significant cost on society—a burden that could be alleviated through appropriate rewetting measures (Loisel and Gallego-Sala, 2022). Additionally, there are growing concerns surrounding climate change projections for the Northern Hemisphere, indicating an expected increase in more frequent extreme precipitation events, along with extended dry periods (Hawcroft et al., 2018; AghaKouchak et al., 2020).

Pristine peatlands function as significant water reservoirs, efficiently storing substantial amounts of water during periods of high rainfall (Acreman and Holden, 2013). As extreme rainfall events are anticipated to become more frequent in the evolving global climate, understanding the role of peatland rewetting in flood mitigation is increasingly vital. Rewetting projects typically involve physical interventions such as ditch-blocking and re-profiling, aiming to increase GWL. Moreover, the blocking of ditches cuts off preferential pathways along open drains, and when combined with pooling behind dams, has the potential to act as a buffer during peak flow events, slowing water release and mitigating the flashiness of the discharge response (Holden, 2006, Holden and Burt, 2003). Therefore, by reducing peak flows, peatland rewetting can also contribute to natural flood management (NFM) by attenuating downstream flow and diminishing flood risk. Furthermore, the reduction of peak flows could play an important role in mitigating further erosion of the peatlands and minimizing sediment production, as well as carbon loss (Shuttleworth et al., 2015).

The effect of peatland rewetting on hydrological responses during rainfall events has received scientific attention over the past decades (Gatis et al., 2023; Goudarzi et al., 2021; Shuttleworth et al., 2019; Menberu et al., 2018; Ketcheson and Price, 2011). Event-based analysis of stream hydrographs, employing various metrics related to hydrograph magnitude and timing, is a common approach for investigating dominant runoff generation processes in catchments and understanding how quickly water is mobilized from the landscape (Ketcheson and Price, 2011, Kirchner et al., 2023; Haque et al., 2022). These response metrics provide valuable insights into catchment storage and release mechanisms (Blume et al., 2007). One widely acknowledged aspect is the impact of rewetting on the event runoff coefficient, which represents the ratio of event runoff depth to event rainfall depth (Evans et al., 1999; Shuttleworth et al., 2019). Therefore, comparing event characteristics before and after rewetting offers



a means to understand hydrological processes and runoff generation mechanisms at the catchment scale,
thereby improving our understanding of flood estimation during extreme events.

A common limitation in the current literature is the predominant focus on event characteristics in natural
or relatively unimpacted catchments, with few studies addressing rewetted peatlands. Additionally, the
extent of hydrological changes due to rewetting is not well understood. Some studies highlight the
positive impact of peatland rewetting on flood moderation (Gatis et al., 2023; Shuttleworth et al., 2019;
Javaheri and Babbar-Sebens, 2014; Beven et al., 2004; Lane et al., 2003; Wilson et al., 2011), but there
are inconsistencies in the extent of flood moderation. For example, Gatis et al. (2023) reported a 49%
reduction in peak storm flow after rewetting, while Shuttleworth et al. (2019) found a 24% reduction in
peak storm flows and a 94% extension in lag times without a change in runoff coefficients. The
challenges in understanding the effects of rewetting at the catchment scale are further underscored by
the inherent high spatial variability of peatland hydrology and physical characteristics (Evans et al.,
1999). The apparent discrepancies in study outcomes, coupled with significant variations among
different research sites, highlight the importance of addressing this through further in-depth
investigations.

Moreover, a recent meta-analysis conducted by Bring et al. (2020) has brought attention to a noteworthy
knowledge gap in understanding the impact of rewetting on GWL changes at different distances from
the intervention. While existing studies have contributed valuable data on the overall hydrological
effects of peatland rewetting, a comprehensive spatial analysis of groundwater changes following
rewetting remains inadequately explored. Despite this shortage, some studies suggest that the impact of
rewetting, especially through ditch blocking, is localized, resulting in more pronounced GWL rise in
close proximity to the ditch (Haapalehto et al., 2014; Wilson et al., 2010; D'Acunha et al., 2018;
Armstrong et al., 2010). Our prior study (Karimi et al., 2024) in the same catchment site investigated
the overall effect of rewetting on hydrological functioning and reported a significant rise in GWL post-
rewetting. However, a thorough examination of groundwater changes at varying distances from the
ditch, considering its crucial role in discharge regulation, is essential to enhance our mechanistic
understanding of flow generation after rewetting. Without such monitoring, the estimation and
extrapolation of discharge responses across landscape extents become more uncertain. Therefore, a
more detailed spatial analysis of GWL changes is crucial for those involved in managing these
peatlands.

Addressing the variability in peatland hydrological responses is essential for developing effective
strategies in peatland management, especially given the evolving trend in climate. Despite a growing
body of research, persistent uncertainties exist regarding the effectiveness of rewetting across diverse
sites and the mechanisms governing peatland recovery (Ketcheson and Price, 2011; Holden et al., 2004).



Additionally, the post hoc nature of monitoring at many restoration sites, driven by projects prioritizing
the speed and cost-effectiveness of restoration work over the scientific robustness of monitoring,
exacerbates these challenges. These time-constrained, funding-driven limitations results in a shortage
of landscape-scale, controlled, or long-term monitoring studies, hindering the development of
comprehensive insights into the long-term effects of peat restoration. The need for more extensive and
sustained research is therefore paramount to fill these critical gaps and advance our understanding of
peatland dynamics in the face of environmental changes.

In Sweden, peatlands cover approximately 65,600 km2 (16% of the Swedish land area) and are
predominantly located within boreal regions (Franzen et al., 2012; Montanarella et al., 2006). The
historical practice of draining peatlands began in the early 18th century for agricultural purposes and
later in the 19th century for forestry, resulting in the excavation of over 1 million km of ditches,
primarily dug by hand to facilitate forestry (Laudon et al., 2022). Consequently, the rewetting of
degraded peatlands in Sweden has become a pressing priority to enhance the hydrological functions of
these ecosystems (Bring et al., 2022). As a response, several national programs for peatland rewetting
have emerged, with a primary emphasis on reintroducing essential ecosystem services, notably flood
control. In a significant move, in 2018, 27 million euros was allocated to facilitate peatland rewetting
in Sweden. However, the scientific underpinning supporting the desired outcomes of peatland rewetting
is still largely lacking.

Given that there have been inconsistent reports in the literature on the extent to which rewetted peatlands
will affect hydrological functioning, particularly with regards to NFM, we build on methods used to
examine the effect of pristine peatlands on flood attenuation (Karimi et al. 2023) to that of rewetting's
impact on hydrological functioning. We used a hydro-climate data set comprised of one-year pre- and
three year post-rewetting and incorporate two control catchments to ensure the robustness of our
findings. The primary objective of this paper was to test whether peatland rewetting has any NFM effect.
We hypothesized that rewetting leads to a reduction in peak flow, runoff coefficient, Hydrograph Shape
Index (HSI), and an increase in lag time, resulting in a generally less flashy hydrograph. Moreover, as
GWL is an important indicator of the amount of water stored in the peatland and the effect of the
rewetting, we asked how far from the ditch GWL was increased by the ditch blocking. We hypothesized
that the areas closest to the ditch would increase the most of any distance from the ditch compared to
the areas farther away from the blocked ditch.



## 2. Materials and methods

### 2.1 Study sites

This study took place in the Trollberget Experimental Area (TEA), situated approximately 50 km northwest of Umeå (TEA; 64.181550N, 19.835378E) (Fig. 1). The TEA's peatland is an oligotrophic minerogenic fen dominated by *Sphagnum* spp., complemented by sparse sedges, dwarf shrubs, and slow-growing individual Scots pine (*Pinus sylvestris*). The underlying soils consist mainly of humic podzol, with some drier areas featuring Humu-ferric podzol and wetter regions comprising Histosols. Peat depth is on average 2.41 m (Laudon et al., 2023). The climate of the area is classified as cold temperate humid, characterized by a mean average temperature of 2.4°C and annual precipitation of 623 mm (approximately 30% as snow), based on data collected from 1980 to 2020 at the nearby Svartberget Climate Station (Laudon et al., 2021).

The peatland at TEA was drained by digging ditches in the early 1920s primarily for forestry purposes. Prior to rewetting, the bulk density of the drained peatland varied between 0.05 to 0.13 g/cm3 within the top 55 cm of the peat profile. The bulk density generally increased with distance from the central ditch and with peat depth (Casselgård, 2020). TEA includes one large peatland, "Stormyr" that drains in two directions. Thus, the monitoring is conducted using v-notch weirs at the outlets of the two catchments, R1 and R2 (Fig. 1). In November 2020, trees within the peatland were cut and the peatland was rewetted using 20-ton crawling excavators to block the drainage ditches, utilizing on-site peat and trees to fill in the man-made ditches that had been present for approximately 100 years to re-establish wetter conditions (Laudon et al., 2021). As a result of these efforts, 34% of the ditches in the 47 ha catchment of R1 and 16% of the ditches in the 60 ha catchment of R2 have been blocked.

### 2.2 The Degerö Stormyr

This study leveraged available data from a nearby natural fen, Degerö Stormyr (273-ha catchment), located in the Kulbäcksliden Research Infrastructure (KRI) (64.182029N, 19.556543E) to serve as the control for the rewetted peatland at TEA (R1 and R2). Degerö Stormyr is characterized as an acidic, oligotrophic, minerogenic, mixed mire system. This intensively studied peatland complex exhibits varying vegetation compositions, predominantly featuring Sphagnum moss and sedges. The depth of the peat has an average thickness of 3-4 m (Noumonvi et al., 2023). The bulk density of the peatland varied between 0.02 to 0.06 g/cm3 within the top 34 cm of the peat profile (Fig. 2 in Casselgård, 2020). The climate of the site is characterized as cold, temperate, and humid, with a mean annual precipitation of 645 mm and a mean annual temperature of +3°C, based on a 30-year average (1991–2020).




### 2.3 C4 (Kallkälsmyren)

The second control catchment, C4 (Kallkälsmyren), situated within the Krycklan Catchment Study
(KCS) (64.260722N, 19.770339E). C4 is a nutrient-poor, minerogenic fen located approximately 10
km from the rewetted catchment. It encompasses an area of 18 ha, with 40% covered by peatlands and
the remainder by forest (Laudon et al., 2021). Similar to TEA, the climate is characterized as a cold
temperate humid type with persistent snow cover during the winter season. The peat vegetation cover
is dominated by *Sphagnum* spp.

### 2.4 Data collection

At the TEA, GWLwere measured between 2019 and 2023 at an hourly resolution using 30 dipwells.
Half of these dipwells were continuously monitored for GWL using data loggers (Solinst Levelogger
5), while the remaining were manually measured every two weeks during the snow-free season.
Dipwells were distributed along 5 transects. Each transect consisted of 6 wells with increasing distances
of approximately 10, 50 and 100 m from the main ditch (Fig. 1). For the Degerö Stormyr control site,
GWL data for the corresponding period were obtained from the ICOS database (www.icos-
sweden.se/data). Due to technical issues with the groundwater loggers, no groundwater data for recent
years was available for the C4 control catchment in the Krycklan Catchment Study.
The discharge data at two TEA mire outlets was collected between 2019 and 2023 at an hourly
resolution using 90 degree sharp-crested V-notches with connected data loggers for continuous water
level measurements (Tru-track). Automatic observations were not possible year-round as there was no
heating in place, which limited data collection during the winter low flow periods. Frequent manual
water level measurements were made to calibrate automatic water level data, and stage-discharge
relationships were defined using manual flow gauging. Specific discharge (discharge per unit catchment
area) was calculated using catchment areas derived from the Deterministic 8 (D8) algorithm based on
a $2 \times 2$ m resolution DEM in which we first burned the ditches into the DEM to the depth of 0.5 m
(Whitebox GAT 3.3) (Laudon et al., 2021). For this study, we utilized discharge data from the C4
control site due to its proximity to the rewetted site. At C4, the outlet is equipped with a V-notch weir
situated within a heated dam house, facilitating continuous stage height monitoring year-round.
Discharge measurements and calibrations followed the same protocol and interval as those implemented
at TEA (Laudon et al. 2021).
Rainfall data were acquired from a reference climate station at Svartberget Research Station
(64.244376N, 19.766378E, 225m a.s.l) (Laudon et al., 2021). Rainfall measurements were logged every
10 minutes using a tipping-bucket (ARG 100, Campbell Scientific, USA). The climate station is integral



to the reference climate monitoring program at Vindeln experimental forests, adhering to the WMO
standard for meteorological measurements (Karlsen et al., 2019).
## 2.5 GWL analysis
First, the hourly groundwater data were examined for outliers, and any gaps were filled using the
Generalized Extreme Studentized Deviate (ESD) filter (Rosner, 1975). The algorithm processes a time-
series dataset by calculating a rolling mean and standard deviation with a window size of 6 hours.
Outliers were identified by comparing each data point to the moving average, and values exceeding the
3-standard deviation threshold were identified as outliers and subsequently removed from the dataset.
Subsequently, the data were gap-filled using the Spline interpolation method, an advanced form of
interpolation that utilizes piecewise polynomial functions to estimate data between two known points.
The data were aggregated to daily time scales. For our analysis we used the GWL data from 1st of June
to the end of October as our study focused on rainfall events; before this date, precipitation often occurs
as snow and dipwells could be frozen. For each catchment R1 and R2, the GWL data were averaged,
and pairwise comparisons test were conducted to assess if there were any significant differences
between pre-rewetting and multiple post-rewetting years. As the data were not normally distributed and
we were interested in the distribution of the data and not the means, the non-parametric Wilcoxon tests
were used. Then, a Bonferroni-Holm correction was applied to adjust for multiple comparisons. The
differences were considered significant when $p < 0.05$. Moreover, to examine the impact of rewetting
on GWL at all distances from the main ditch, data were disaggregated based on distances of 10, 50, and
100 m to the main ditch. It is noteworthy that the dipwells were also located near other side ditches,
indicating a potential limitation in the study design.
## 2.6 Rainfall-runoff events detection
As a first step, we segmented the 2020–2023 summer–autumn precipitation record into distinct rainfall
events using the inter-event time definition (IETD) via the IETD R package (Duque, 2020). The IETD
establishes a minimum dry period between independent rainfall events as a criterion for grouping them.
To distinguish independent rainfall events from continuous precipitation, we set a minimum threshold
of 0.1 mm h$^{-1}$ at the start of an event. Events were considered distinct if they were separated by at least
12 hours without rainfall. The methodology for identifying runoff events was based on the framework
outlined by Luscombe (2014) and was further adapted to the specific characteristics of our study area.
Runoff events were defined as periods during which the observed discharge exhibited significant
deviations from the baseflow. This was achieved by considering both the rate of change in discharge
and its magnitude. Peaks in discharge exceeding predefined thresholds were classified as runoff events.
To pair the rainfall and runoff events, rainfall events were matched with the runoff events that followed
within a specified time window. A final, visual inspection of the time series with detected events was





used to quality control these data and ensure that all significant rainfall and flow events were extracted
from the dataset.
2.7 Flood mitigation effects
To evaluate the Natural Flood Mitigation effect of peatland rewetting and determine its impact, we
employed a set of response metrics to characterize hydrologic responses during events following the
rewetting process. These response metrics include event duration, rainfall volume, peak flow, runoff
coefficient, lag time, and Hydrograph Shape Index (HSI). We calculated these response metrics for both
the rewetted and control sites. The selection of these response metrics was based on their widespread
use in hydrological comparison studies (Edokpa et al., 2022; Wilson et al., 2011). Peak flow response
was computed as the maximum discharge observed during each event. Runoff coefficient was
determined as the ratio of total event runoff to total event rainfall. Lag time calculated as the time
between peak rainfall and peak discharge in each event. HSI, defined as the ratio of peak storm
discharge to total storm discharge, provides a straightforward measure of the overall hydrograph shape
(Shuttleworth et al., 2019). The response metrics for the rewetted catchment R2 and the control site
were derived using the start and end times of rainfall-runoff events identified at R1 catchment.
2.8 Statistical analyses
The statistical design used in this study focuses on the BACI approach (before-after and control-impact)
as used previously in hydrological studies (Laudon et al., 2023; Holden et al., 2017; Shuttleworth et al.,
2019; Menberu et al., 2018). We standardized the response metrics derived from the two catchments
(R1 and R2) of the rewetted site against the control catchment (treatment minus control) to distinguish
responses resulting from rewetting treatment from natural variation, changes over time and seasons.
Due to variations in the frequency of events between the pre- and post-rewetting periods, and the non-
normal distribution of response metrics, a non-parametric test was employed. Specifically, the
Wilcoxon test was conducted to investigate statistically significant changes in the distribution of data
for each catchment (R1 and R2) of the rewetted site before and after rewetting, with a focus on
understanding the extremes, rather than solely examining means (Shuttleworth et al., 2019).
Significance was determined at $p < 0.05$. Additionally, we aggregated all years post-rewetting together
due to the highly variable number of events occurring during each year post-rewetting. Statistical
analysis was undertaken in R version 4.1.2. (R Core Team, 2021) with data processing, summary
statistics and plotting undertaken using the R package Tidyverse (Wickham, 2017).



























**Figure 1.** Trollberget Experimental Area (TEA) catchments with monitoring locations (A). Pink circles show the locations of the outlets of the catchment areas for R1 and R2 (weir locations) of the rewetted peatland. Green circles designate groundwater dipwells. Aerial view of rewetted peatland with GWL monitoring transects visible as white lines, summer 2021(B). (Photo by Andreas Palmén)






## 3. Results

### 3.1 The impact of rewetting on GWL variation

Peatland rewetting has led to a significant increase in GWL at the two catchments (R1 and R2) of the rewetted site compared to the control site (Fig. 2a). The relative difference in GWL between the rewetted and control sites (treatment minus control) at varying distances to the ditch also showed a significant decreased after rewetting (Fig. 2b). Interestingly, this impact demonstrated variability depending on the distance from the ditch, with wells located closest to the ditch showing a more pronounced response compared to those farther away. Prior to rewetting, the median GWL was lowest next to the ditch (−228 mm) and highest at the furthest distance away (−174 mm). Furthermore, GWL exhibited greater variability in the middle of the transect (50 m from the ditch), reaching a minimum of 507 mm from the ground. After rewetting, the largest median GWL change was observed at a distance of 10 meters, with an increase of 119 mm. This was followed by a median 91 mm increase at a distance of 100 meters and a median 62 mm increase at a distance of 50 meters. The median GWL at the control sites was roughly the same during the pre and post-rewetting periods (-79 and -78 mm, respectively) (Table 1).



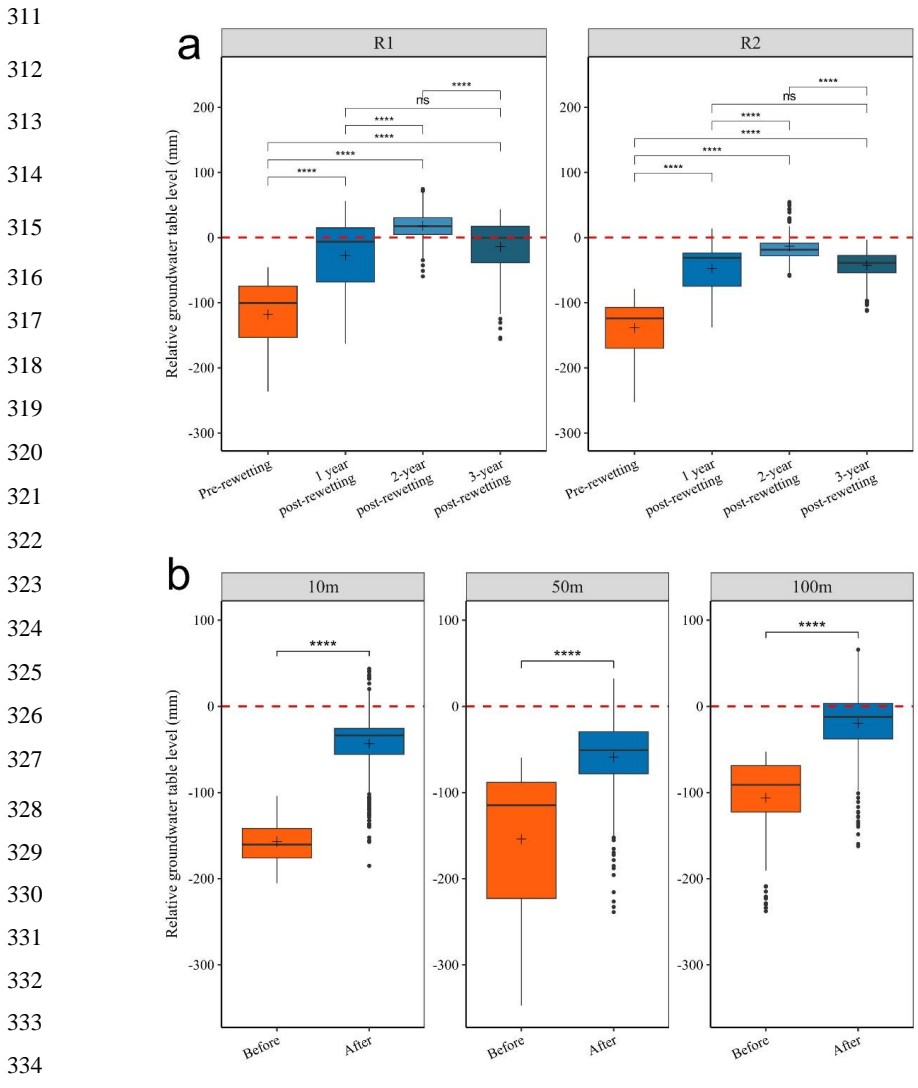

**Figure 2.** a) Relative difference (treatment-control) in GWL based on daily data gathered between June to October in the years 2020 (pre-rewetting) and 2021, 2022 and 2023 (3 years post-rewetting) regardless of distance to ditch. b) Relative difference in GWL based on varying distances to the main ditch; all years post-rewetting are combined (sample sizes for pre-rewetting and post-rewetting were 153 and 428, respectively). The red dashed line indicates the value of the control site; positive values indicate that the value is greater at the rewetted site than at the control, while negative values indicate the opposite. The box plots show the minimum, first quartile, median, third quartile, and maximum, with outliers as dots. The stars indicate the levels of significance difference between the marked comparisons as determined using a Wilcoxon test (****$p \leq 0.0001$).





**Table 1.** Median, minimum (min), maximum (max) and 5th-95th quantile of GWL change pre- and post-
rewetting for different distances to the ditch and the control site.

| | Distance | Median(mm) | Min (mm) | Max (mm) | 5th- 95th quantile (mm) |
|---|---|---|---|---|---|
| PRE-REWETTING | 10 m | -228 | -364 | -120 | 194 |
| | 50 m | -190 | -507 | -60 | 370 |
| | 100 m | -174 | -416 | -44 | 304 |
| | Control | -79 | -186 | 8.5 | 156 |
| POST-REWETTING | 10 m | -108 | -272 | -33 | 197 |
| | 50 m | -127 | -366 | -30 | 233 |
| | 100 m | -83 | -341 | 5.4 | 240 |
| | Control | -78 | -234 | 2.7 | 171 |



## 3.2 The impact of rewetting on runoff responses

Based on the response at R1, 17 rainfall-runoff events before and 30 events after rewetting were
extracted and analyzed (Fig. 3). The impact of rewetting on runoff responses during rainfall-runoff
events is depicted through examples of event-scale hydrographs (Fig. 4, Table 2), illustrating the
variation in discharge response across control and the two catchments (R1 and R2) of the rewetted site
for different event sizes and antecedent GWL conditions, both pre-and post-rewetting periods. In the
pre-rewetting period, despite the control site having the shallowest GWL at -15 mm, it exhibited the
lowest peak flow of 0.29 mm/h. In contrast, rewetted site R1, with an antecedent GWL of -82, reached
a peak of 0.93 mm/h. One and two years after rewetting, R1 still had the highest peak at 0.71 and 0.61,
respectively, while the rewetted catchment R2 showed similarities to the control site. However, three
years after rewetting, although R1 had the shallowest antecedent GWL at -5.15 mm, the peak flow was
almost half of the peak in the control catchment (0.14 and 0.26, respectively).









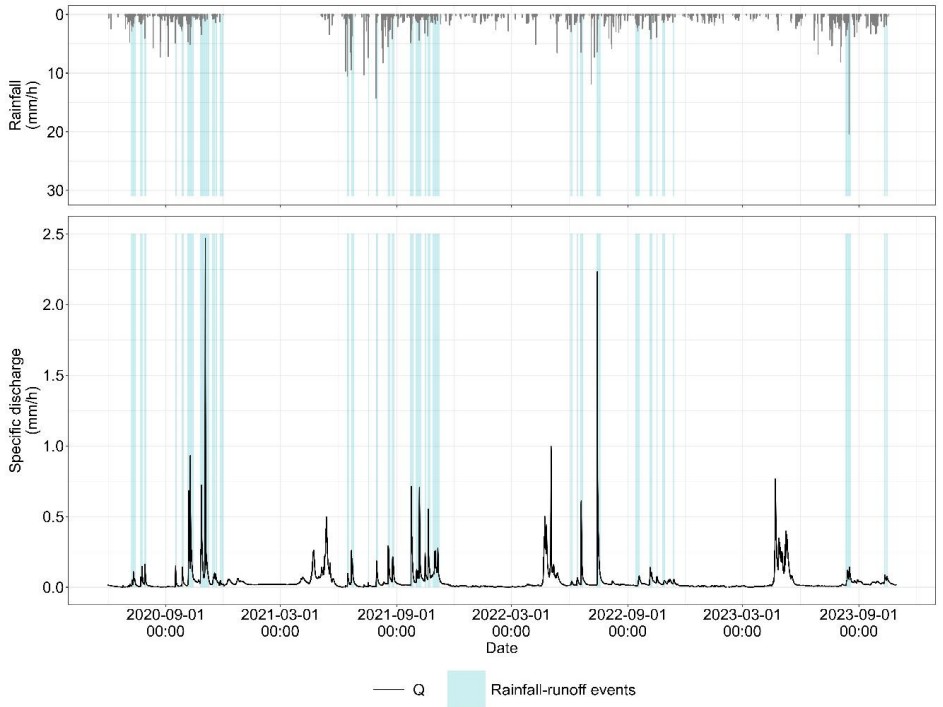


**Figure 3.** Identified rainfall-runoff events using discharge measured at the rewetted
catchment R1 across the entire study period.
















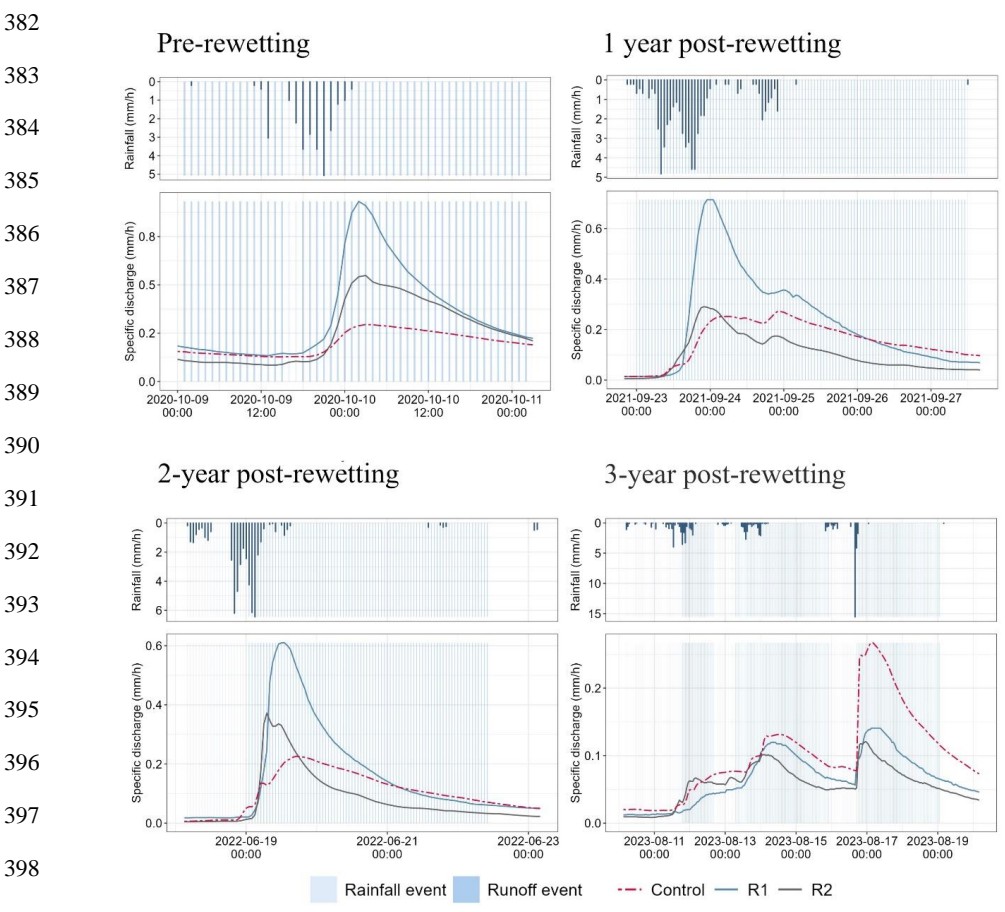

**Figure 4.** Examples of runoff responses of control and the two catchments (R1 and R2) of the rewetted site during rainfall-runoff events for each of the four pre- and post-rewetting years.






**Table 2**. Characteristics of the 4 rainfall-runoff events shown in Figure 5 for the rewetted (R1 and R2) and control sites during the pre- and post-rewetting years.

| | Site | Total rain (mm) | Peak flow (mm/h) | Antecedent GWL (mm) |
|---|---|---|---|---|
| Pre-rewetting | | 37 | | |
| | Control | | 0.29 | -15 |
| | R1 | | 0.93 | -82 |
| | R2 | | 0.54 | -102 |
| 1 year post-rewetting | | 63 | | |
| | Control | | 0.27 | -35 |
| | R1 | | 0.71 | -24. |
| | R2 | | 0.29 | -85 |
| 2-years post-rewetting | | 53 | | |
| | Control | | 0.22 | -47 |
| | R1 | | 0.61 | -34 |
| | R2 | | 0.37 | -57 |
| 3-years post-rewetting | | 69 | | |
| | Control | | 0.26 | -19 |
| | R1 | | 0.14 | -5.1 |
| | R2 | | 0.12 | -40 |

### 3.3 Flood mitigation effects of rewetting

The magnitude of the effects of peatland rewetting was investigated for 47 rainfall-runoff events (17 events before rewetting and 30 events after rewetting) to test if the rewetting's effects were significant under a larger number of events. Storm magnitudes ranged between 5 and 50 mm in total precipitation before rewetting, and 2.3 and 63 mm after rewetting. The relative differences between the two catchments (R1 and R2) of the rewetted site and control sites (rewetted minus control) for each metric are shown in Fig. 5.

The analysis of rainfall-runoff events revealed a reduction in relative peak flow at the two catchments (R1 and R2) of the rewetted site following rewetting (Fig. 5a). However, the reduction was significant only at R1. Specifically, the median peak flow at R1 decreased from 0.14 to 0.10 mm/h post-rewetting. In contrast, at R2, there was an increase from 0.04 to 0.08 mm/h post-rewetting. Interestingly, the control site experienced a rise in median peak flow from 0.05 to 0.12 mm/h during the post-rewetting period.





Moreover, the median runoff coefficient in the two catchments (R1 and R2) of the rewetted site showed
an increase from 0.36 to 0.4 and from 0.14 to 0.20 at R1 and R2, respectively, after rewetting. The
runoff coefficient at the control site increased from 0.17 before rewetting to 0.40 after rewetting.
Relative to the control site, both restored sites, R1 and R2, experienced a decline in runoff coefficients
during the post-rewetting phase. Notably, this reduction was statistically significant solely at R1
($p < 0.01$ and $p < 0.05$, respectively) (Fig. 5b).
After rewetting, the median lag time in the two catchments (R1 and R2) of the rewetted site decreased
by 0.5 and 7 hours, reaching 15 and 10 hours for R1 and R2, respectively, compared to the pre-rewetting
values of 14 and 17 hours. In contrast, the control catchment exhibited an increase in median lag time
from 14 to 23 hours during the post-rewetting period. However, pairwise test results indicated that there
was no statistically significant change at both rewetted catchments (R1 and R2) following rewetting
(Fig. 5c).
The median HSI values for both catchments (R1 and R2) of the rewetted site and control sites decreased
after the rewetting period, shifting from 0.023 to 0.021, 0.034 to 0.025, and 0.027 to 0.026 at control,
R1, and R2, respectively (Fig. 5d). The effect of rewetting in reducing HSI was significant only at R1
($p < 0.0001$). Prior to rewetting, the relative HSI at R1 was 0.012, and after rewetting, it decreased to
0.003. The relative HSI also experienced a decline at R2, dropping from 0.006 pre-rewetting to 0.004
after rewetting. However, this decrease was not statistically significant.

















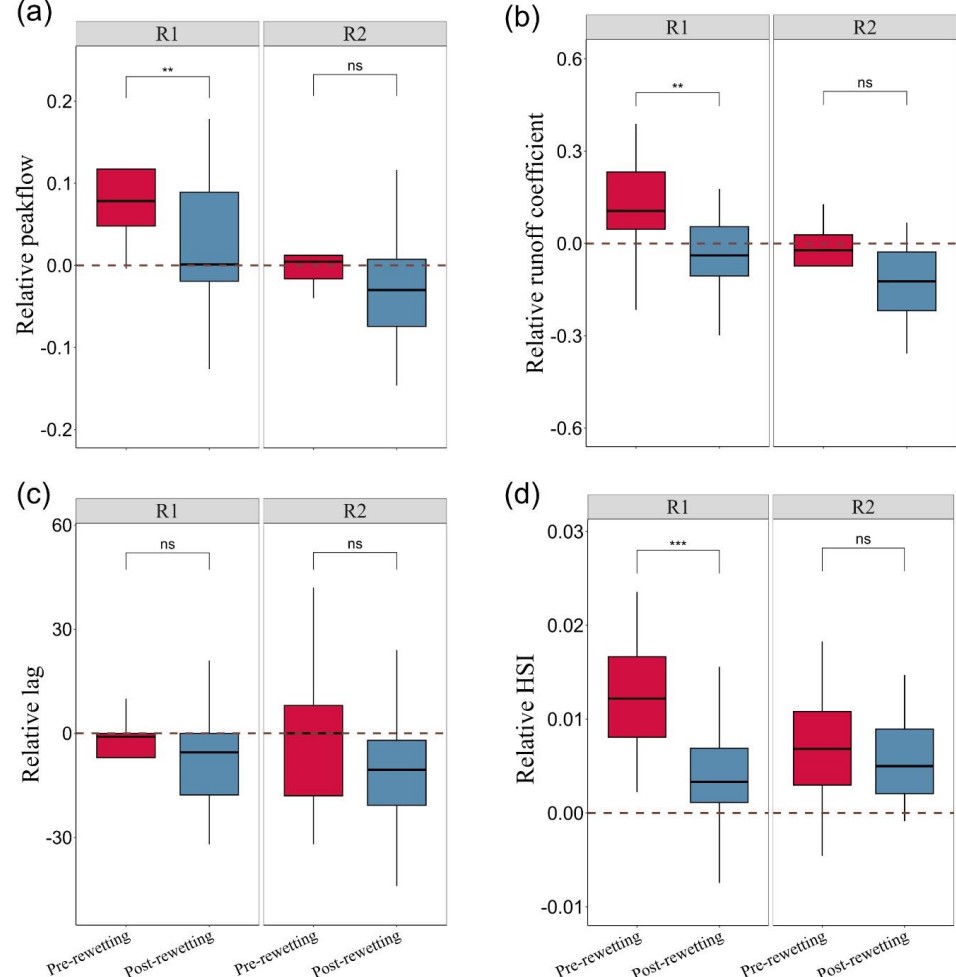


**Figure 5**. Differences between the rewetted and control sites pre- and the combined three years of post-
rewetting period for (a) peak flow, (b) runoff coefficient, (c) lag time, and (d) Hydrograph Shape Index
(HSI). The relative difference was computed as treatment minus control and the red dashed line indicates
the value of the control site; thus positive values indicate that the solute is greater at the treatment site
than at the control site, while negative values indicate the opposite. The box plots show the minimum,
first quartile, median, third quartile, and maximum, with outliers as points. The stars indicate the levels
of significance in Wilcoxon test (**$p \leq 0.01$; ***$p \leq 0.001$; "ns" denotes not significant.).





## 4. Discussion

Considering the diverse characteristics of peatlands in the boreal biome, our results show a generally positive impact of peatland rewetting on GWL, runoff responses during rain storms, and the effectiveness of restoration efforts in mitigating floods on nutrient-poor minerogenic mires, which are one of the most common peatland types in Fennoscandia.

### 4.1 The impact of rewetting on groundwater table level (GWL)

Using the BACI experimental approach, we evaluated how closely the mean GWL position of the rewetted sites matched that of the pristine control site after ditch-blocking of both R1 and R2. This aligns broadly with several other studies (Shuttleworth et al., 2019; Armstrong et al., 2022; Howie et al., 2009; Haapalehto et al., 2014; Dixon et al., 2014; Menberu et al., 2016; Soomets et al., 2023) that found that rewetting raised GWL to near pristine levels. Our results also revealed that the median GWL at R1 closely resembled that of the control site after rewetting. However, at R2, the median GWL remained slightly lower post-rewetting. This difference may be attributed to the presence of shrubs and sparse tree cover (higher water uptake) on the mire at R2, as well as a lower proportion of blocked ditches within the catchment. Additionally, our results addressed a gap in the existing literature by examining the spatial variability of GWL recovery at different distances from the ditch, a factor largely neglected in prior research, particularly within the context of boreal ecosystems (Bring et al., 2022). We demonstrated that the GWL increase after rewetting was spatially variable but occurred at all distances from the main ditch. Contrary to the assertion made by Bring et al. (2022) that the impact of rewetting on GWL diminishes with increasing distance from the main ditch, our results reveal a significant increase in GWL at all distances after rewetting. Furthermore, the inclination of GWL toward the ditch before rewetting was reduced after rewetting.

Similar to our result, Haapalehto et al. (2014), found in a study conducted in southern Finland, that ditch-blocking raised the GWL up to 800 mm in the vicinity of the ditch. They observed a lower GWL at 0 m from the ditch compared to 10 m and 15 m before rewetting. Following rewetting, no significant differences were noted between the locations. However, in our study, significant differences persisted even after rewetting. Similarly, in eastern Finland, Laine et al. (2011) investigated the influence of ditch-blocking on GWL and they found that during the period from August to October 2007, filling the ditches led to a rapid rise in the GWL, reaching the same level as the pristine fens, both next to the ditch and in the middle of the strip (peat profile between ditches). Conversely, some studies found no significant impact of distance to the ditch. For example, Wilson et al. (2010) demonstrated that blocking raised the GWL downslope of ditches by approximately 20 mm, but they found that the distance did not significantly affect GWL after blocking. However, their plot that shows the mean GWL at different distances to the ditch indicated that the inclination toward the ditch remained after rewetting. The





515 difference in GWL between 10 m and 30 m from the ditch was 30 mm, while at our study site, the

516 difference between 10 m and 50 m was 15 mm. In a similar study, Holden et al. (2017) conducted

517 research in a blanket peatland in the UK and, through strict ANOVA analysis, found no significant

518 effect based on the distance from the blocked ditch. However, they observed that the midpoint between

519 the transects had the highest GWL compared to the wells closest to the ditch.

520 On the other hand, some studies showed that the effects of rewetting may be localized, occurring mainly

521 in close proximity to the ditch (Armstrong et al., 2010; Cooper et al., 2014). For example, in a study in

522 Southwestern British Columbia, Howie et al. (2009) examined the impact of ditch-blocking on GWL

523 at different distances from the ditch. They found that GWL responded to ditch-blocking only locally,

524 within a short distance from the blocked ditch (20 m). This localized effect observed in their study could

525 be attributed to the intense degradation of their peatland, combined with extensive peat extraction,

526 resulting in significant alterations in vegetation from mosses to shrubs and trees. Furthermore, the

527 extensive drying of the peatland, coupled with shrinkage and subsidence of the peat, led to a reduction

528 in hydraulic conductivity, possibly hindering the effectiveness of restoration efforts in reversing the

529 impacts of drainage.

530 Additionally, there have been instances where rewetting did not result in a rise in groundwater levels

531 (GWL), even in proximity to the blocked ditch, as demonstrated by Williamson et al. (2017). They

532 conducted a study assessing the impact of ditch-blocking on aeration depth. Their investigation revealed

533 that historical peat compaction and subsidence within a 4–5 meter zone adjacent to the ditch effectively

534 reduced the peat surface to the GWL after drainage, making the peatland less responsive to rewetting

535 due to pre-existing saturation. However, as they mentioned, this phenomenon was mainly observed in

536 temperate lowland and tropical peat sites, whereas studies in boreal peatlands drained for forestry have

537 yielded different outcomes. Overall, as hypothesized, the most significant changes occurred in the

538 vicinity of the ditch and the GWL inclination decreased between distances after rewetting. This detailed

539 spatial monitoring of GWL at different distances to ditch was necessary to ensure that all of the locations

540 in the mire extents had undergone rewetting as part of a major rewetting initiative and any observed

541 differences in event runoff responses could be attributed to changes in GWL and water storage within

542 the peatland. Furthermore, our data serves as a valuable resource for peatland managers, helping them

543 to gain a better understanding of site-specific hydrological changes and the associated ecosystem

544 services that result from the rewetting of peatlands, rather than relying on sporadic measurements of

545 GWL at a few points within the mire.

546

547

548





### 4.2 The impact of rewetting on runoff responses

Event-based analysis of discharge responses is crucial, as relying solely on daily discharge analysis may not offer a detailed temporal scale to precisely identify changes in the rapid response of discharge to precipitation, including the lag time to peak flow. For instance, examining the hourly hydrograph revealed that, although discharge responses at R1 exhibited flashier characteristics with higher peaks compared to those at R2, the lag time to peak at R2 post-rewetting was notably shorter than at R1. This discrepancy could possibly be attributed to a lower proportion of blocked ditches at R2. However, the scarcity of continuous, prolonged datasets from rewetted peatlands, particularly in Sweden, poses a significant challenge in conducting comprehensive comparisons across various peatland sizes and rewetting durations, as most rewetting projects have only recently commenced. Therefore, a more extended period of post-rewetting monitoring is necessary to fully understand how the discharge patterns of drained peatlands evolve after rewetting.

### 4.3 Flood mitigation effects of rewetting

Rewetting resulted in a significant reduction in event peak flow response at R1. The decrease in the peak flow was not significant at R2. By reducing peak flows, peatland rewetting delivers natural flood management (NFM) by attenuating downstream flow and reducing flood risk. Our findings align with the results observed in Wilson et al. (2011), where they showed peak flow hydrographs from ditches with considerable change after rewetting, with lower peak flow rates, less runoff and less of the rainwater being released during the event. In contrast, Shantz and Price (2006) evaluated the hydrological characteristics of a restored peatland in Quebec, Canada and observed higher discharge peaks during summer at the restored site compared to the control site, attributing it to wetter antecedent conditions and faster drainage response following rainfall. However, our research reveals that despite observing a rise in GWL after rewetting, rewetted peatlands can exhibit less flashy flood responses and offer improved retention of rainfall. This suggests that contrary to conclusions drawn in many previous studies (Holden, 2005; Holden et al., 2004) about reduced potential storage capacity, the rewetted peatlands in our study exhibit more controlled and resilient hydrological behavior.

Runoff coefficient is another key indicator for flood mitigation and corresponds to catchment storage capacity. Our results showed that reduction in runoff coefficient was significant at R1, showing less runoff being exported with rainfall events after rewetting, but again, this reduction was not significant at R2. The effect of peatland rewetting on reducing runoff coefficient has been reported in many studies (Shantz and Price, 2006; Wilson et al., 2011; Gunn and Walker, 2000; Ketcheson and Price, 2011). Ketcheson and Price (2011) specifically investigated the impact of ditch-blocking on an abandoned cutover peatland in Canada over a period of two years before and one-year after rewetting. Their findings highlighted a substantial reduction in the runoff coefficient as the most significant hydrological


effect of peatland rewetting. However, caution in interpreting these results due to the potential influence
of the relatively short time series during which the peatland was undergoing filling. In contrast,
Shuttleworth et al. (2019) reported conflicting results in their investigation using a BACI experimental
design in the South Pennines, UK. Their study on blanket peat restoration on hillslopes, including
revegetation and gully blocking, did not reveal any significant impact on the storm runoff coefficient
for either treatment, but this is likely because these peatlands are located on slopes while our rewetted
sites are at the outlet of the basin. In another study by Menberu et al. (2018), they examined the impact
of rewetting on hydrological responses within seven small peat-dominated catchments in Finland. They
employed three different approaches to extract hydrological events. Interestingly, the runoff coefficients
calculated using two of the approaches, which were most similar to our methodology, showed higher
values 3 and 4 years after restoration in the restored catchment compared to the control areas. They
suggested that this increase could be attributed to the declining efficiency of the dams, resulting in
increased runoff over time.

While an increasing lag time traditionally serves as a positive indicator for flood modification, contrary
to expectations, the lag time between the initiation of a rainfall event and the peak discharge decreased
after rewetting. However, it's important to note that this decrease, while observed, did not reach
statistical significance.  This result is in line with findings from other studies (Wilson et al., 2011; Gatis
et al., 2023; Ketcheson and Price, 2011). One plausible explanation for this paradox lies in the research
conducted by Wallage and Holden (2011), who explored the impact of different peatland management
strategies (specifically, drained and restored) on GWL, near-surface macropore flow, and saturated
hydraulic conductivity in a blanket peat headwater catchment in northern England. Interestingly, the
researchers found that the rewetted peatlands exhibited higher surface hydraulic conductivity compared
to their intact counterparts. The upper peat layers in the rewetted areas allowed for greater water
movement as throughflow, in contrast to the intact site, thereby contributing to a decrease in lag time.
In contrast to our observations, Shuttleworth et al. (2019) reported a 106% increase in lag time through
revegetation and gully blocking. However, it is not obvious how the effect of gully blocking would have
been without revegetation measures, as the increase in lag time might be attributed to the heightened
surface roughness provided by the newly established vegetation.

Hydrograph Shape Index (HSI), serving as an indicator of system flashiness, exhibited a notable
decrease at catchment R1 following the rewetting process while this reduction was not significant at
R2. This reduction aligns with findings from other studies (Shuttleworth et al., 2019; Wilson et al.,
2011; Gatis et al., 2023). For example, Gatis et al. (2023) investigated the impact of rewetting a blanket
bog on hydrograph shape using General Additive Models (GAM) and reported a 68% decrease in the
mean gradient of the hydrograph rising limb. Wilson et al. (2011) conducted a study on hydrograph
changes in ditches and small streams within the Lake Vyrnwy catchment in mid Wales. Their research





focused on the impact of drain blocking in blanket peat, revealing significant decreases in peak flow and hydrograph flashiness after the implementation of drain blocking measures. Shuttleworth et al. (2019) also reported a 37% reduction in HSI after gully blocking and revegetation of a blanket peatland. In contrast, a study by Regensburg et al. (2021) examined the impact of peatland restoration through pipe outlet blocking on the hydrological functioning of a blanket peatland in Northern England. Their study, which included the calculation of a Response Index similar to HSI, found no direct impact on any of the event response metrics based on their Before-After-Control-Impact (BACI) analysis. The lack of immediate impact could be attributed to the steeper gradients in their study site. However, their post-rewetting monitoring, spanning a relatively short period of six months, may not capture the long-term effects, suggesting that flood moderation might occur in the more extended period after restoration efforts.

The significant decreases in peak flow, runoff coefficient and HSI observed at R1, compared to the non-significant changes at R2, can be attributed to several factors. Firstly, the BACI analysis indicated that, prior to rewetting, R1 had much flashier hydrological responses than R2. In contrast, R2's responses were already more similar to the control site, suggesting a less potential changes post-rewetting. Additionally, a smaller portion of catchment R2 was restored, which could mean that the overall water storage at R2 remains lower than at R1. Consequently, water may still drain more quickly at R2, leading to less noticeable impacts from the rewetting efforts. Moreover, the diverse responses observed in flood response characteristics, both in our study and in other investigations, raises questions regarding the overall effectiveness of peatland rewetting. While it appears successful in reducing peak flow, runoff coefficient, and overall flashiness of hydrographs (as shown by HSI), the evidence suggests it might not be as effective in increasing lag time from peak rainfall to peak flow occurrence. This limitation could potentially be attributed to the need for new peat formation. However, a crucial question regarding the duration of these effects and the time necessary for lag time recovery remains unanswered. The effectiveness of ditch-blocking in flood moderation is influenced by various factors, including the initial condition of a drained peatland, the extent of peat degradation, and changes in its properties (Menberu et al., 2016). Furthermore, there may be a delayed effect in the peatland's response to ditch-blocking, and the corresponding flood mitigation may progressively change over time in the years following the blocking of ditches due to changes in peat properties and vegetation cover. Moreover, our three-year monitoring period post-rewetting, yet longer than many other studies, offers limited insight into the impact of rewetting on flood moderation under extreme storm events, especially in more severe future climate conditions. Therefore, further monitoring is required to understand the influence of restoration practices on peatland hydrological functioning.





## 5. Conclusion

In this study, we employed the Before-After-Control-Impact (BACI) design to assess the impact of peatland rewetting on flood control in a nutrient-poor boreal minerogenic fen in northern Sweden. Continuous hourly hydrometric data spanning one year before (2020) and three years after rewetting (2021, 2022, and 2023) were utilized for this evaluation. Additionally, groundwater level (GWL) data from various distances to the ditch were provided to demonstrate the entire areas within the peatland affected by rewetting, which is essential for capturing storm responses arising from the rewetting process. Analysis of the discharge time series indicated that the effect of rewetting on flow moderation is not as fast as rising GWL. This gradual and evolving process of peatland hydrological functioning due to a long history of peat compaction and decomposition and subsequent re-establishment of peat-forming vegetation after rewetting emphasizes the importance of sustained long-term monitoring to fully understand the outcomes of rewetting. Moreover, the findings indicated that peatland rewetting has the potential for flood mitigation and even mitigated rainfall events better than the pristine site in some cases. However, significant changes were only observed at one of the outlets, R1. This was supported by reductions in peak flow, runoff coefficient, and less flashy hydrograph responses (HSI). However, the results showed that peatland rewetting would not necessarily increase the lag time between the peak of a rainfall event and peak discharge. Nevertheless, uncertainties persist in our understanding of the Natural Flood Management (NFM) contribution of peatland rewetting over longer timescales or during large historical flood events. Therefore, we emphasize the significance of long-term monitoring combined with hydrological modeling to determine whether the flood attenuation function of peatlands remains consistently applicable under future climate change, where floods are expected to become more frequent and extreme.

## Code and data availability

All data used in this study are freely available. The discharge data can be obtained from https://data.fieldsites.se/portal/ . The groundwater table level data up to October 2023 are available from the corresponding author. The original R codes for extracting rainfall-runoff events are available from Gatis et al. (2023) at https://ore.exeter.ac.uk/repository/handle/10871/134028.

## Financial support

The TEA infrastructure was initiated and co-funded by the European Union GRIP on LIFE IP project (LIFE16IPE SE009 GRIP) led by the Västerbotten Administration Board and Swedish Forest Agency, with additional financial infrastructure and research support from The Kempe Foundation and the Swedish Research Council Formas grants (2018-00723 (to EMH), 2018-02780 (to HL), 2020-01372 (to HL), 2021-02114 (to HL), as well as by the Knut and Alice Wallenberg (Grants 2018.0259 and


2023.0245). The KCS/KFI infrastructure and long-term data collection have been funded by The
Swedish Research Council VR (SITES, grant number 2021-00164).

## Acknowledgements

We would like to thank all the skilled and dedicated field personnel at the Svartberget research station.

## Competing interests

The authors declare that they have no conflict of interest.

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
