# Peer review of "Does peatland rewetting mitigate flooding from extreme"

_Hydrology and Earth System Sciences, 2024_

## Author Response (AR1)

**Reviewer 1**

**General comments**

This manuscript assesses the impact of peatland rewetting on peatland flood mitigation in a nutrient-poor boreal minerogenic fen in northern Sweden. The study used an impressive four year long dataset of hourly hydrometric and GWL data, which the authors utilized with valid and clearly described scientific methods. The study was well-designed as it included data both before and after rewetting from both the treated and control sites. This is a novel study that contains important results concerning rewetting effects on peatland hydrology and it fits perfectly within the scope of HESS. However, the discussion section suffers from text bloat and loss of focus due to unnecessarily long summaries of the referenced papers, which has replaced the proper discussion of some of the results reported in this manuscript. Also, more details or at least rephrasing the descriptions of the study sites are needed. After some work, I think this manuscript will be suitable for publication in HESS.

Author's response:

*We thank Referee #1 for the overall positive comments as well as providing constructive criticism and suggestions that improved the next version of the manuscript. Regarding the discussion section, we addressed both the general and specific comments and eliminated the unnecessarily long summaries. Our responses to all the comments are listed below in the order they appear.*

*Abstract*

You don't write that you had two catchments and that the results differed between them. Currently, you only summarize the results from the catchment where you observed significant changes and present that as the sole result of your study. I think it should be mentioned that you also had the other catchment and there you did not observe significant changes. You can briefly mention why it is not important in terms of the conclusions of the study if that is the case. My understanding is that catchment R2 was already quite similar to pristine peatland conditions and because of that, the changes were not significant there.

Author's response:

*We agree with Referee #1. We now mention our results from the other catchment (R2) and why we think it is still valid to say that rewetting improved flood mitigation by adding the phrase:*

*"Yet, our results showed that the effectiveness of ditch-blocking in flood moderation was strongly influenced by the initial condition and catchment percent of restoration, as one of our two rewetted peatlands did not show significant change attributed to it being already similar to the pristine site, suggesting less treatment effect; and the other catchment, with higher restoration percentage, had a better response to treatment."*

*Introduction*

The introduction addresses well the relevant scientific questions and gives a nice background for the study. However, I think the introduction is a little bit on the longer side. The first three paragraphs were really nice and clearly structured, but after that, there were parts which were dragging a bit and I think they could be written more concisely. For example, you could consider if the paragraph starting on L100 is really necessary or if it can be shortened or parts of it moved into some other part of this section.

In my opinion, the paragraph starting on L112 is not in the correct place. I think it should be much closer to the beginning of the section, for example, between the pristine peatland and rewetting paragraphs (L57). This paragraph is very Sweden-specific, so that should be led already on previous paragraphs somehow as they concentrate mostly on the European scale and suddenly having a paragraph written solely on Sweden's point-of-view is a little too much of a contrast.

Author's response:

*We thank Referee #1 for the nice initial comment. Regarding the changes suggested, we agree and moved the paragraph starting on L112 to L57 (now L63), therefore starting from a European and moving to a national context. Furthermore, also as suggested, we moved the important parts of the paragraph starting at L100 to other paragraphs which improved the flow of the introduction considerably.*

*Methods*

The utilized methods are well-described and scientifically valid and I have no major comments on those. However, the site descriptions need some clarification (see the "specific comments" below).

Add the company names when reporting the used instrumentation.

Author's response:

*Regarding the specific comments about the study site, they are all very good comments that we addressed either as suggested or by fixing what was confusing. Please check in the specific comments below how we specifically addressed them. Finally, as also suggested by Referee #2, we improved how we report the instrumentation by adding company and country of origin.*

*Results*

The results were clearly presented and I enjoyed reading them.

Check that you use the hyphen and en-dash properly throughout the manuscript. For example, in 3.1. you use both of them in the same use cases (L303, L308), which does not make sense.

Fix the unit formatting: mm/h --> mm h-1.

Author's response:

*We are glad Referee #1 enjoyed them, and thank Referee #1 for mentioning it. We fixed the formatting to mm h$^{-1}$ and the use of hyphen and dash in all places needed.*

*Discussion*

This section is the one needing the most amount of work. There is a lot of summarizing of the referenced papers, which unnecessarily bloats the text. I'm not saying you should not compare your results to other studies at all, but I would like to see more answers to questions containing "why?". Why did something happen/didn't happen (you can use your hypotheses as a base), why your results were different/similar to other sites' results, etc.? Now most of the discussion is summarizing what previous studies have done and then you compare your results to those sometimes with just one short sentence. This is not the point of the discussion section. See more details in "specific comments".

The last paragraph was the one I was waiting to read for the whole discussion section. This is what the discussion should contain (here: "Why the results from the two catchments differ from each other?")!

On L220 you say :"It is noteworthy that the dipwells were also located near other side ditches, indicating a potential limitation in the study design." I did not see any discussion about how this may have affected the results. I think it should be added.

Author's response:

*Following the comments of Referee #1 (but also Referee #2) regarding our discussion, we completely restructured it. This was done by removing the over summarization of other papers and instead focusing on trying to answer why our rewetted sites responded the way they did, however not without comparing our results with others. Overall, we believe our discussion dramatically improved, offering a proper discussion of some of the results reported in this manuscript.*

*We changed the discussion as follows:*

[revised manuscript text omitted]

Nice conclusions. I have two small suggestions.

This is the same comment I gave about the abstract as well. I think it should be briefly mentioned why the changes at R2 were not significant. Otherwise, one might get a feeling that you had two catchments

and only one had experienced significant changes after rewetting, but you are making generalization based only on that catchment that experienced significant changes and ignoring/hiding the results from the other catchment.

Author's response:

*We thank Referee #1 for pointing this out. We completely agree with the suggestion, hence we added to the conclusion:*

*"Significant changes were only observed at one of the outlets, R1, and although R2 did show positive changes they were not significant. These differences seem to be attributed to (1) a higher percentage of catchment restoration resulting in a better response to treatment and (2) R2's responses being already similar to the pristine site, suggesting less potential changes post-rewetting."*

It would be nice if you could condense the first three sentences describing what you did in this study. It currently takes about one-third of this section, which should be about the conclusions of the study. I suggest removing the last one of these three sentences and moving the mention of GWL to the second sentence.

Author's response:

*We thank Referee #1 for the suggestion. We agree and we need to focus on the conclusions rather than summarize again what we did in the study. Therefore, we eliminated those three sentences and went directly to the conclusions. The conclusion now reads:*

*"Our results showed that the effect of rewetting on flow moderation from rainfall events is not as fast as rising GWL. This gradual and evolving process of peatland hydrological functioning due to a long history of peat compaction and decomposition and subsequent re-establishment of peat-forming vegetation after rewetting emphasizes the importance of sustained long-term monitoring to fully understand the outcomes of rewetting. Moreover, the findings indicated that peatland rewetting has the potential for flood mitigation and even, in some cases, mitigated runoff from rainfall events better than the pristine site. This was supported by reductions in peak flow, runoff coefficient, and less flashy hydrograph responses (HSI). However, the results showed that peatland rewetting would not necessarily increase the lag time between the peak of a rainfall event and peak discharge. Significant changes were only observed at one of the outlets, R1, and although R2 did show positive changes they were not significant. These differences seem to be attributed to (1) a higher percentage of catchment restoration resulting in a better response to treatment and (2) R2's responses being already similar to the pristine site, suggesting less potential changes post-rewetting. Nevertheless, uncertainties remain in our*

*understanding of the contribution of peatland rewetting to NFM over longer timescales or during large historical flood events. Therefore, we emphasize the significance of long-term monitoring combined with hydrological modeling to determine whether the flood attenuation function of peatlands remains consistently applicable under future climate change, where floods are expected to become more frequent and extreme."*

**Specific comments / Technical corrections**

L4: "Agricutural" --> "Agricultural"

[Author's response]{.underline}:

*We fixed the typing error in our university name.*

L14: Do you want to say that rewetting peatlands can mitigate the effects of flooding caused by extreme rainfall events? I read this sentence so that rewetting can mitigate specific weather conditions (here: when it is raining a lot). This same comment applies also to the title of the paper. Consider rephrasing.

[Author's response]{.underline}:

*We thank Referee #1 for this comment, we now see the confusion. We changed the title to: "Does peatland rewetting mitigate flooding from extreme rainfall events?" as suggested by referee #2 and we changed L14 to: "To assess whether rewetting peatlands can mitigate flooding from extreme rainfall events and ensure water security in a future climate,"*

L33: 15% of the boreal land/surface area or something else? Please clarify.

[Author's response]{.underline}:

*Correct, we changed it to "They encompass 15% of the boreal land surface area"*

L38: "…with more than half of the peatlands estimated to have been lost…" Do you mean half of the original peatland area or what? Also, I wouldn't say that the peatlands are lost in some of the activities you mentioned, but rather transformed as the peat is still there in the soil. I suggest using "pristine peatlands" instead of just "peatlands" to clarify what you want to say.

[Author's response]{.underline}:

*We changed the wording to be more accurate: "with more than half of the total pristine peatlands in Europe estimated to have been impacted by drainage for agriculture, forestry, or peat extraction"*

L40: Could you give some examples of the ecosystem services that they cannot sustain?

Author's response:

*Yes, it is important to highlight the important ecosystem services. We added "ecosystem services such as such as reduction of carbon losses or buffering extreme hydrological events"*

L75-L77: I don't think it is an inconsistency that both studies you reference here reported a reduction in peak storm flows (+ only other observed extension in lag times), although a different amount. Or what do you mean by these "inconsistencies", could you clarify or use different wording? You use the word "inconsistent" later in the introduction as well, check that those also make sense.

Author's response:

*Referee #1 is right, both studies agree in a reduction in peak storm flow, however we are trying to refer to the extent of this reduction and how it seems to not be universal. We change it to "discrepancies in the extent of flood moderation."*

L140-L141: Is this the vegetation/tree stand situation at the site before rewetting or after? You should be more clear about the description of the site in terms of before and after rewetting.

Author's response:

*Referee #1 is absolutely right. We now clearly state that this information was taken before restoration. This also clarified the question about Scots pine posed ahead.*

L147: The peatland was drained for forestry purposes, so the site was an actual forest before rewetting? GWL seems to have been quite high (section 3.1) before rewetting, so it was not likely a dense forest with tall trees? Is this a typical result of drainage in Sweden? On L141 you write that there were "individual Scots Pine", which gives the impression that there were only some pine trees there and there. Could you clarify this? Did you measure tree stand characteristics (diameter, height, basal area, etc.) at the site before rewetting? It would be nice to have that as supplementary information. Were all the trees cut during the rewetting as you say on L152? Fig 1b shows that there are some trees left in the eastern part of the site. Is that the case (then correct what you say on L152) or was the picture taken before the cuttings were finished?

Author's response:

*Again, we agree. It is a very straight forward situation that needs to be clear. Furthermore, while restructuring the discussion we thought of importance to mention an old open-water pool that re-*

*appeared after restoration. Hence we also mentioned it in the study site section. We rewrote this section and will read as:*

*"The peatland at TEA was drained by manual ditch-digging in the early 1920s primarily for forestry purposes, however because of nutrient limitation the peatland remained unproductive with sparse tree cover (Laudon et al., 2023). The peatland is divided into two catchments draining in two directions, referred to here as R1 and R2, with drainage areas of 33 and 60 ha, respectively (Figure 1). Thus, our monitoring was conducted using v-notch weirs at the individual outlets of the two catchments. Both catchments are similar in topography and vegetation, however R1 previously had an open-water pond shown on historic maps pre-drainage. In the 1930s, the uplands of the peatland were also drained leading to 1152 m of ditches in R1 and 5189 m of ditches in R2 (Laudon, Sponseller, & Bishop, 2021). In November 2020, the peatland was rewetted by filling and blocking the ditches in the peatland, not the upland, and as a result of these efforts, 59% of the ditches of R1 and 16% of the ditches of R2 were blocked. The ditches were filled using peat from the site with additional dams built at regular intervals using the tree logs harvested from the site. The logs were placed horizontally but perpendicular to the ditch, except at the two outlet locations where the logs were inserted vertically into the peat and layered additionally with geotextile. To protect the soil characteristics as much as possible, the heavy machinery (i.e., excavators) used moveable log mat while moving on the mire. Additionally, the sparse tree cover that grew on the peatland was cut to reduce evapotranspiration and complement the ditch blocking (Karimi et al., 2024). Finally, after restoration, the open-water pond at R1 re-appeared ca 100 m upstream of the sampling location (Figure 1b)."*

L152: Were the two catchments similar in terms of the site characteristics you report in this paragraph? Please clarify.

Author's response:

*It is one peatland with similar characteristics draining in two directions. However, the catchment area is different and we now propose to mention that R1 catchment previously had an open-water pond shown on historic maps pre-drainage and that after restoration re-appeared. Hence, we would clearly state that it is one peatland with similar characteristics draining in two directions and would mention the old open-water pond in R1. The sentence would now read:*

*"The peatland is divided into two catchments draining in two directions, referred to here as R1 and R2, with drainage areas of 33 and 60 ha, respectively (Figure 1). Thus, our monitoring was conducted using v-notch weirs at the individual outlets of the two catchments. Both catchments are similar in topography and vegetation, however R1 previously had an open-water pond shown on historic maps pre-drainage."*

L158: How near was Degerö to the Trollberget? Add the distance to the text.

Author's response:

*Degerö is approximately 24km from the TEA. We added that information.*

L166: What is the source of this data? Why did you calculate the mean temperature and precipitation for a different period for Degerö and Trollberget (L144-L146) sites? Maybe the correct question is why do you report different numbers for these sites when they are located so close to each other? I think that only one set of climatological statistics is enough.

Author's response:

*We agree. We deleted this second section of climatological information.*

L177: Add space after GWL.

Author's response:

*We added space.*

L180: "5" --> "five, "6" --> "six"

Author's response:

*We changed them.*

L181: How many GWL loggers were there in Degerö?

Author's response:

*They are four wells. We added this information.*

L188-L189: How frequent were those manual water level measurements?

Author's response:

*The measurements were taken twice a month. We added this information also.*

L193: So you did not use/have discharge data from the Degerö site? If so, I think it should be mentioned explicitly.

Author's response:

*We mentioned that we only use discharge data from C4, due to proximity to the rewetted site.*

L199: Add space between "225" and "m".

Author's response:

*We added it.*

L212: "before this date" --> "outside this period"

Author's response:

*We changed as suggested.*

L232: What were these "predefined thresholds"?

Author's response:

*To correctly answer this question we expanded our method to show step by step what we did. This is how we addressed it:*

*"Runoff events were defined as periods during which the observed discharge exhibited significant deviations from the baseflow. Rainfall events were matched with the runoff events that followed within a specified time window (12.5h). We calculated rolling quantiles for this time window (12.5 h) at the 30th and 95th percentile (Q30th and Q95th respectively). A rolling quantile for the 70th percentile for a one month period is also calculated (MQ90). Where (Q95th – Q30th) > MQ90, the flow is considered to be elevated and any fluctuation in flow is driven by precipitation; therefore measured Q is used (Gatis et al., 2023; Puttock, Graham, Ashe, Luscombe, & Brazier, 2021). A final, visual inspection of the time series with detected events was used to quality control these data and ensure that all significant rainfall and flow events were extracted from the dataset."*

Figure 1: What kind of stream/ditch is the one at the bottom of R1? Should that have been considered in this study? Does the figure contain all the ditches at the site (related to my later comment)?

Author's response:

*We agree with Referee # 1, the ditch showed in the lower part of figure A was not taken into consideration in the study. The line shown is from the Swedish National database of streams and are not complete, especially when ditches are concerned. We deleted it from the map to avoid confusions. Furthermore, we added to the upper figure (A) all the ditches in the peatland that were blocked and all the unblocked ditches in the catchment area.*

Figure 1: The distances between the transect lines in A and B figures do not match. What I mean is that in the lower figure, it looks like there are two groups of two transects and one sole transect, but in the upper figure there are two sole transects and a group of three transects. Why is that?

Author's response:

*We see our mistake and the source of confusion. The lower figure (B) is mainly focusing on the mire, what can be seen as white lines are the boardwalks which are not all directly related to the groundwater wells transects. For example, the first two boardwalks are linked to an individual groundwater transect and the other boardwalks are associated with greenhouse gas chamber measurements. Furthermore, the further west groundwater transect of our study does not have a boardwalk, thus it seems missing in the lower figure. Therefore, to avoid confusion we will correct the legend in the figure to identify the boardwalks.*

L297: Remove "has"

Author's response:

*Did it.*

L300: "decreased" --> "decrease"

Author's response:

*We made the correction.*

L303: Was the GWL significantly different on different distances from the ditch before and after the rewetting?

Author's response:

*Yes, before and after rewetting the GWL were significantly different among all distances from ditch, thus we did not think it was important to mention this as there was no change after rewetting.*

Figure 2: Why did you combine the post-rewetting data into one in 2b and not show the years separately as you did in 2a?

Author's response:

*Great question, we really thought about this. We tried a couple of versions, however the year-by-year version with all the stats seemed too cluttered and was difficult to follow. We thought it seemed straight forward to just show the before-after graph therefore simplifying the figure and hence communicate*

*better the message. However, we mentioned that changes in the different distances from ditch followed the same pattern through time:*

*"Finally, GWL in the three different distances from ditch significantly increased in the first year after rewetting, significantly increased further in the second year and significantly decreased in the third year, however still being significantly higher than the pre-rewetting and first year post-rewetting GWL."*

Table 1: Rounding to two significant digits would be accurate enough in this table.

Author's response:

*We prefered to just round up the decimals but keep the precision.*

L360, L361, L364: Add the missing units.

Author's response:

*Fixed as suggested.*

L362: "… R2 showed similarities to the control site." Add a reference to Table 2.

Author's response:

*Absolutely, we did this.*

Figure 3: Add what "Q" is into the figure caption.

Author's response:

*Great suggestion, we changed the figure legend.*

Figure 4: It would be nice to have the same y-axes for the specific discharge subplots, so their year-to-year variation would be easier to see.

Author's response:

*Although we agree that changing the y-axis in all specific discharge subplots would make it easier to see the change from year-to-year, we wanted to focus on showing the change in the peakflow between control and the peatland catchments. Hence, we maintained the y-axis difference. However, we added a note in the figure legend highlighting that "scales for the y-axes show different magnitudes of specific discharge".*

Table 2:

In the caption: "4" --> "four". Should it be "Figure 4" instead of "Figure 5"?

I'm not sure if this table is necessary and it could be combined with Fig. 4. One can see the peak flows from Fig. 4 already, so showing that again here in the table is not needed. Also, total rainfall can be added as a number within the rainfall subplots in Fig. 4. However, antecedent GWLs are a bit trickier. You could add a separate subplot for GWL similarly as you have the rainfall. Then you could add the antecedent GWL as a number somewhere within the GWL plot if you think that it needs to be highlighted separately. Alternatively, you could add GWL data as a second y-axis for the rainfall subplot.

Author's response:

*First, great catch from Referee #1 on the number of figure referenced. We will change it. Furthermore, we agree that the table maybe is not necessary, and we thank for the suggestions. Therefore, we decided that we could move the table to supplementary information.*

L431: "2.3" --> "2". I don't think you need to be that accurate here considering that you are talking about precipitation.

Author's response:

*Agree. We left it at 2mm.*

L440: Remove "Moreover".

Author's response:

*We did.*

L445: I'm confused, what do those p-values represent? R1 is $p < 0.01$ and R2 is $p < 0.05$? But you just wrote that the reduction was statistically significant solely at R1, but if the p-value was smaller than 0.05 for R2, that would mean the reduction was significant for R2 as well, right? Please fix/clarify. Also, add into the brackets a clarification of what site the p-value represents (ex. R1: $p < 0.01$ and R2: …).

Author's response:

*Sorry about that mistake. P value for R2 was supposed to have $p > 0.05$. Since we already established in the methods section that significant p values will be $p < 0.05$ and to try and make it easier for the reader, we only wrote the p value of R1 as it is the only significant one ($p < 0.01$).*

L447: I don't think it is worth mentioning the pre- and post-rewetting lags and their decrease in the same sentence. It should be enough to say the change in the lag time and then either pre- or post-rewetting lag times. The easiest fix is to remove the rest of the sentence after "respectively".

Author's response:

We thank *Referee #1 for the suggestion, we did that.*

L484: What makes you think you can make such a generalization based on results from just this one site? Also, your observed changes were significant only at one of the two catchments you measured.

Author's response:

We a*gree with Referee #1, we added a sentence establishing our different results and another drawing attention to the diverse characteristics of peatlands. This paragraph reads as:*

*"Despite significant interest in peatland rewetting, there is limited research on its effects on hydrological functioning and the scale of these impacts. We found that peatland rewetting on nutrient-poor minerogenic fens, one of the most common peatland types in Fennoscandia, was generally positive for use in Natural Flood Management. Rewetting has begun to affect GWL, runoff responses during rain storms, and flood mitigation (although this latter was only in one of two study catchments) and move these hydrological characteristics more towards pristine conditions storing more water in the peatland. However, special attention should be on the diverse characteristics of peatlands in the boreal biome before generalizing the effect of peatland rewetting on hydrological functioning."*

L487-L488: I think this first sentence is missing the result, which you are comparing to other studies in the next sentence.

Author's response:

*We restructured this first sentence as follows:*

*"Using the BACI experimental approach, we found that the mean GWL position of the rewetted sites rose to near pristine levels of our control site after ditch-blocking of both R1 and R2 and that the rise of GWL was generally rapid after restoration"*

L493: Lower than what?

Author's response:

*Lower than the control. It is clearer this way:*

*"Furthermore, our results also revealed that the median GWL at R1 closely resembled that of the control site after rewetting. However, the median GWL remained slightly lower at R2, compared to the control, after rewetting."*

L493-L494: Related to my previous comment on the tree stand data. Here it would be nice to have some hard data about tree stand and vegetation on which to base these conclusions.

**Author's response**:

*In the method section, we added information about basal area of tree canopy in the restored area. Specifically:*

*"Prior to rewetting the peatland was dominated by Sphagnum spp., complemented by sparse sedges, dwarf shrubs, and sparse tree canopy (basal area = 2.6 m2 ha−1) of slow-growing Scots pine (Pinus sylvestris)."*

L498 & L500: "We demonstrated that the GWL increase after rewetting was spatially variable but occurred at all distances from the main ditch." & "…, our results reveal a significant increase in GWL at all distances after rewetting." Unnecessary repetition.

**Author's response**:

*This unnecessary repetition was eliminated with the new restructured discussion.*

L504: The 800 mm rise in GWL in Haapalehto et al. (2014), in my opinion, was not similar to your result. You wrote on L305-L306 that the largest rise in GWL was 119 mm, which is much smaller compared to the referenced paper.

**Author's response**:

*These sentence was removed in the new restructured discussion.*

L503-L519: This paragraph contains too many details from the referenced papers and very little discussion about your results. Comparing to previous studies is ok, but there is no need to have an "extensive" summary of each referenced paper.

**Author's response**:

*We agree, after the restructuring of the discussion we eliminated all over summarization of the referenced papers.*

L520-L529: How is this related to your study? There is no mention of that. This paragraph is just a summary of three previously published papers. This is not the purpose of the discussion section.

:

*Again, we agree and this type of paragraphs disappeared after the restructuring of the discussion.*

L537: There should be a paragraph break here before "Overall". The text before it is just another summary of a previously published study. The latter half is better as you address your hypothesis, but I think it should be done earlier in the discussion maybe at the start of the second paragraph. After doing that, you would compare your results to other studies and address why they are different/similar to your study. I'm pretty sure that you can reduce the length of the discussion about GWL change in different distances to about half of what it now is just by reordering the text and removing unnecessary extensive summarizing of the referenced papers.

:

*The restructured discussion presents a better section about GWL change.*

L555: Why was a lower proportion of ditches blocked in R2 compared to R1? Also, looking at the blue and red lines in Fig. 1a, it seems like 99% of the ditches were blocked in R2, which is more than in R1 and conflicts with your statement. Either you meant that a lower proportion of the ditches were blocked in R1 compared to R2 or you are not showing all the ditches in Fig. 1a. If you are not showing all the ditches in the figure, you should add them.

:

We did not show all the ditches in the previous maps. *We added to the upper figure (A) all the ditches in the peatland that were blocked and all the unblocked ditches in the catchment area, hence showing clearly why the percentage of blocked ditches is higher in R1 compared to R2.*

L556: "particularly in Sweden" So there are studies outside Sweden? Why you are not mentioning those? I understand that such datasets are very limited, however from this choice of words I get a feeling that you know that there are other studies, but for some reason, you are not referencing those.

:

*We were more thinking about, for example, the lack of information of fens compared to blanket bogs, we changed that in the discussion and reads as:*

*"However, the scarcity of continuous, prolonged datasets from rewetted peatlands, particularly for boreal minerogenic fens, poses a significant challenge in conducting comprehensive comparisons across various peatland sizes, types, and rewetting durations, as most rewetting projects have only recently commenced. Therefore, a more extended period of post-rewetting monitoring is necessary to fully understand how the discharge patterns of drained peatlands evolve after rewetting."*

L584: "However, caution in interpreting these results…" Some words are missing here. "… caution should be taken in when…"?

Author's response:

*We added "is needed" and the sentence changed to:*

*"However, caution is needed in interpreting some of these results due to the potential influence of the relatively short time series during which the peatland could have been still undergoing filling (Ketcheson & Price, 2011) or an eventual increase in the runoff coefficient due to a declining efficiency of the ditch blocking (Menberu et al., 2018)."*

L642: "the evidence suggests…" What evidence? Your study? Or some other studies (add references in that case)?

Author's response:

*We meant evidence based in our results. Hence, we changed the sentence to:*

*"While it appears successful in reducing peak flow, runoff coefficient, and overall flashiness of hydrographs (as shown by HSI), our results suggests it might not be as effective in increasing lag time from peak rainfall to peak flow occurrence"*

Citation: https://doi.org/10.5194/hess-2024-158-RC1

**Reviewer 2**

**General comments**

This paper used a BACI (before-after and control-impact) approach to investigate how peatland restoration (ditch blocking) effected the hydrology at their study site in northern Sweden. The analysis was largely event-based, and compared how the two restored catchments responded to rainfall events against how the control catchment responded. The experiment was well designed and the analysis generally well explained/justified.

The paper is generally well written, if a little long in places. With some cutting down and improvement of the discussion, and minor clarifications to the methods and results, I think this paper will be suitable for publication in HESS, for which it is well within scope.

Author's response:

*We thank Referee #2 for the positive comments as well as providing constructive suggestions that improved the next version of the manuscript. As also mentioned by Referee #1, we addressed the long summaries of the referenced papers and improve the discussion. Our responses to all the comments are listed below in the order they appear.*

**Specific comments**

*Title*

Consider rephrasing slightly. How about; "Does peatland rewetting mitigate flooding from extreme rainfall events?"

Author's response:

*Absolutely, referee #1 also suggested to rephrase the title. We followed Referee #2 suggestion.*

*Introduction*

No specific comments/concerns.

Author's response:

*Great!*

*Materials and methods*

Could you mention more about how the efficacy of the rewetting was quantified? Was it just through measuring GWL, or were there any aerial surveys done?

[Author's response]():

*The efficacy of rewetting is solely based on the increase of GWL on the extent of the peatland.*

Did the excavators affect the bulk density near the surface? Any measurements of this?

[Author's response]():

*Unfortunately we do not have these new measurement yet. However, we added important information on the study site section about how the excavators used moveable log mat in order to protect the soil characteristics as much as possible.*

Some care is needed when stating the equipment manufacturers (notes in technical corrections).

[Author's response]():

*We thank Referee #2 for these comments. We addressed them on the technical corrections.*

*Results*

Results were easy to follow. Figures presented nicely.

[Author's response]():

*We thank Referee #2.*

*Discussion*

The discussion was difficult to follow at times due to its length. There is too much detail from other studies, and not enough on the findings from this study. For example, at present, each paragraph in section 4.3 (except the last starting on L633) generally follows a formula of one sentence on your findings, followed by a summary of the literature. The paragraph starting on L633 is much more engaging as it puts your results in context throughout, and tries to explain them. Coming back to your findings throughout each paragraph of the discussion will make the whole thing flow much better, in my opinion.

[Author's response:]()

*After Referee #2 comments and Referee #1´s comments regarding our discussion, we completely restructured it – see an entirely new discussion in Referee #1 (based on both reviewers).*

*Conclusions*

Conclusions are succinct and appropriate.

Author's response:

*We thank Referee #2 for this comment.*

**Technical corrections**

L49 first use of GWL – write in full. It is later written as groundwater table level (L486) and groundwater level (L530). Choose one or the other.

Author's response:

*Great catch from Referee #2! We wrote in full and change them all to GWL.*

L75 I wouldn't class these as inconsistencies, necessarily. They're more like a range. Lots of factors will have differed between these studies (e.g. catchment characteristics, magnitude of studied rainfall events).

Author's response:

*We agree, also a comment from referee #1. We changed it to:*

*"Some studies highlight the positive impact of peatland rewetting on flood moderation with a reduction of peak storm flow (Gatis et al., 2023; Javaheri & Babbar-Sebens, 2014; Lane et al., 2003; Shuttleworth et al., 2019; Wilson et al., 2011), however there are discrepancies in the extent of the flood moderation"*

L117 functions = functioning?

Author's response:

*We agree, it we changed it to "functioning" as we are referring to the overall systems dynamics.*

L133 This sentence is a little tricky to follow. Change to "We hypothesized that the areas closest to the ditch would increase more than the areas further away from the blocked ditch".

Author's response:

*Great suggestion from Referee #2! We changed as suggested.*

L177 Insert space between GWL and were

Author's response:

*We did.*

L178 where is Solinst from?

Author's response:

*Canada, we added the country of origin in the name.*

L187 believe this is TruTrack. Add country.

Author's response:

*Changed and add the country.*

L192 first use of DEM – write in full.

Author's response:

*Wrote in full.*

L200 …using a tipping bucket [rain gauge]

Author's response:

*We agree with Referee #2, we added it afterwards.*

L200 I believe the ARG100 is manufactured by EML, in the UK (though may have been supplied by Campbell Scientific).

Author's response:

*Referee #2 is very impressively right. We fixed that.*

L238 Natural Flood Mitigation is used here, but Natural Flood Management elsewhere (e.g. L564) and more commonly.

Author's response:

*On this sentence it meant to be flood mitigation effect. We changed it to avoid confusion.*

Figure 1. Might be worth highlighting again in caption how a greater proportion of R1 was rewetted, as this could be missed further up.

Author's response:

*Absolutely. We added the percentage of the ditches that were blocked.*

L300 decreased = decrease

Author's response:

*Deleted the "d".*

Figure 5 is solute a typo?

Author's response:

*Yes it was. We changed it to "hydrological response".*

L531 don't need "groundwater levels" before (GWL) here.

Author's response:

*Correct, we just left GWL and we changed it here and in other places thanks to this comments and another comment before.*

L584 …caution [is needed] in interpreting…

Author's response:

*Correct again. We added "is needed" after caution.*

L681 This link is seems to be broken.

Author's response:

*It seems it was temporally down. The link is working now.*

Citation: https://doi.org/10.5194/hess-2024-158-RC2

**Reviewer 3:**

Kindly note that :

Line 681: The link for the Rcode from Gatis et al (2023) is not working: https://ore.exeter.ac.uk/repository/handle/10871/134028.

Author's response:

It seems it was temporally down. The link is working now.

---

## Author Response (AR3)

**Response from Editor:**

Dear Authors,

Following a thorough evaluation by the reviewers, your revised version of the manuscript has been found to show improvements over the original paper. However, the reviewers have also suggested additional changes before the manuscript can be accepted for the final publication in HESS. I concur with the reviewers' assessments and hereby release their comments for your consideration. Kindly submit a revised manuscript and s point-by-point response to these comments. Should you disagree with a particular comment, please provide a detailed explanation. Please note that a possible quick review by the reviewers might be necessary.

> *Author response: We thank the editor for the opportunity to address the reviewers' minor comments. We have now addressed all the comments (in blue italics).*

**Comments from referee #1**

I have reviewed the author's responses to my earlier comments and I find them satisfactory. Especially, the newly written discussion is markedly better than the previous one. Good job on that! I only have some minor/technical comments left, which I hope the authors take into account.

In addition to typos and confusing sentences noted below, I recommend that the authors proofread the discussion section to fix simple grammatical errors (punctuation, articles). Now the text there seems a bit unpolished.

Author's response: *We thank Referee #1 for this positive comment as well as previously providing constructive criticism and suggestions that improved considerably this version of the manuscript. Our responses to all the comments are listed below in the order they appear.*

**Specific comments:**

Abstract is quite long but if it is within the journal guidelines, I'm fine with it.

Author's response: *We checked and it is within the journal guidelines.*

L152-153: "… not the upland…" What do you mean by this? Were there ditches also in the surrounding upland forests? Why would that be?

Author's response: *There are ditches in the mire catchment that were not blocked, only the ones in the mire itself were blocked, as shown in the map (Figure 1). We explained it differently, and now it reads: "the peatland was rewetted by filling and blocking all the ditches in the peatland, whereas ditches in the surrounding none-peat areas were left unmanaged".*

L180: Add comma after "Solinst". Isn't the company's name "Solinst Eureka"?

Author's response: *As we understand, Solinst eureka is the brand for water quality probes, we believe the name for the loggers is only Solinst. We added the coma.*

L190: The instrument model should probably be TruTrack WT-HR. The name of the manufacturer is also missing (Intech Instruments?).

Author's response: *Correct, it is a TruTrack WT-HR and it is Intech instruments, we added the information.*

L239: What is "Q"? Flow rate? It is not defined.

Author's response: *We are referring to discharge, we changed it.*

Figure 1: This figure is much better than the previous one! One small note though. In Fig 1b, the black "R1" and "R2" are difficult to distinguish from the background. It would be a good idea to change the color to something lighter or even white.

Author's response: *Agree, we changed it to white and looks better.*

L381: "…towards pristine conditions storing more water …" Something (a word and/or a comma) is missing in this part of the sentence.

Author's response: *Good suggestion, we re-wrote the sentence and now it reads: "Rewetting has begun to influence GWL, runoff responses during rainstorms, and flood mitigation (though the latter was observed in only one of the two study catchments) while also shifting these hydrological characteristics closer to pristine conditions by increasing water storage in the peatland."*

L409: The beginning of this sentence is confusing. Please rewrite.

Author's response: *We re-wrote the sentence to make it clearer: "After rewetting, our results show a significant increase in GWL at all distances from the ditch, however with spatial variation"*

L428: "…revealed that, discharge…". Remove the comma.

Author's response: *Removed*

L430: "decrease on" --> "decrease in"

Author's response: *Corrected*

L438: "positively effects" --> "positively affects"

Author's response: *Changed*

L447-L452: This summary of the results at the beginning of the paragraph could be shorter. Here are two suggestions: "L447: remove "however as mentioned above, the decrease in the peak flow was not significant at R2" and on L450: "that did not have a significant change" --> "where the decrease in peak flow was not significant".

Author's response: *Thank you for the suggestion. We changed it as suggested.*

L450: "that" --> "which"

Author's response: *Changed with the comment above.*

L452: "in" --> "by"

Author's response: *Changed.*

L454: Remove "of the"

Author's response: *Removed.*

L467: "suggests" --> "suggest"

Author's response: *Removed.*

L478: I think you could finish the sentence here: "at R2." The rest of the sentence is unnecessary repetition.

Author's response: *We believe this repetition emphasized and important part of our results. Hence, we will keep it.*

L480: Move this sentence: "Specifically, a reduction…" to L476 before "Our results showed that…".

Author's response: *Moved.*

L493: "It's" --> "it is"

Author's response: *Changed*

L493-494: If the decrease in lag time was not statistically significant, the lag time did not really decrease, at least based on your data. Therefore, I don't think it makes much sense to explain the (insignificant) decrease in lag time in such a length as is done in this paragraph. Based on that, consider removing the following four sentences on L494-L502 or heavily modifying them and discuss from the point of view that why you did not observe a significant change in the lag time at your site.

Author's response: *We did a combination of removing sentences and modifying the ones left. However, we did wanted to leave the sentence that explicitly says that the decrease was observable but not significant.*

L519: Remove "in" in "…and in other investigations…"

Author's response: *Removed.*

**Comments from referee #2**

This version of the manuscript is an improvement over the first version, the differences between catchment R1 and R2 are easier to understand, and the discussion explains results from this study in more detail than previously.

[Author's response]():

*We thank Referee #2 for this positive comment as well as previously providing constructive criticism and suggestions that improved considerably this version of the manuscript. Our responses to all the comments are listed below in the order they appear.*

**Suggested technical corrections in the new text:**

L39 serves should be serve

[Author's response](): *Removed.*

L138 between 0.05 AND 0.13

[Author's response](): *Changed*

L157 moveable log mats

[Author's response](): *Added.*

L169 between 0.02 AND 0.06

[Author's response](): *Added.*

L239 Puttock et al (check all newly inserted citations, the format has changed from the original version)

[Author's response](): *Great catch. We now, did a thorough check of all citation style.*

L273 while in R2 only 16% of the ditches (were blocked?)

[Author's response](): *Correct, we added "were blocked"*

L382 special attention should be given to the diverse characteristics.

[Author's response](): *Thanks, we corrected the sentence.*

L388 check in-text citation formatting

Author's response: *We are sorry about this mistake. We now, did a thorough check of all citation style*

L463 therefore (reduce) peak flow?

Author's response: *Added*

L467 results suggest

Author's response: *Couldn't find the mistake there.*

L468 check citation formatting

Author's response: *Again, we are sorry about this mistake. We now, did a thorough check of all citation style*

L478 follow

Author's response: *We edited the sentence based on Referee #1 comment.*

L498 allow not allows

Author's response: *Thanks, we corrected it.*